# Ghrelin and Cancer: Examining the Roles of the Ghrelin Axis in Tumor Growth and Progression

**DOI:** 10.3390/biom12040483

**Published:** 2022-03-22

**Authors:** Anuhya S. Kotta, Abigail S. Kelling, Karen A. Corleto, Yuxiang Sun, Erin D. Giles

**Affiliations:** Department of Nutrition, Texas A&M University, College Station, TX 77843, USA; anuhyak@tamu.edu (A.S.K.); kabigail1025@tamu.edu (A.S.K.); karen_c91@tamu.edu (K.A.C.); yuxiang.sun@ag.tamu.edu (Y.S.)

**Keywords:** cancer, ghrelin, In1-ghrelin, GHSR, tumor

## Abstract

Ghrelin, a hormone produced and secreted from the stomach, is prim arily known as an appetite stimulant. Recently, it has emerged as a potential regulator/biomarker of cancer progression. Inconsistent results on this subject make this body of literature difficult to interpret. Here, we attempt to identify commonalities in the relationships between ghrelin and various cancers, and summarize important considerations for future research. The main players in the ghrelin family axis are unacylated ghrelin (UAG), acylated ghrelin (AG), the enzyme ghrelin O-acyltransferase (GOAT), and the growth hormone secretagogue receptor (GHSR). GOAT is responsible for the acylation of ghrelin, after which ghrelin can bind to the functional ghrelin receptor GHSR-1a to initiate the activation cascade. Splice variants of ghrelin also exist, with the most prominent being In1-ghrelin. In this review, we focus primarily on the potential of In1-ghrelin as a biomarker for cancer progression, the unique characteristics of UAG and AG, the importance of the two known receptor variants GHSR-1a and 1b, as well as the possible mechanisms through which the ghrelin axis acts. Further understanding of the role of the ghrelin axis in tumor cell proliferation could lead to the development of novel therapeutic approaches for various cancers.

## 1. Introduction

Ghrelin is a 28 amino acid endogenous peptide hormone, primarily produced and secreted by the stomach. It is commonly called the hunger hormone due to its orexigenic effects [1,2]. In addition to its role in nutrient sensing, appetite, and meal initiation, ghrelin has emerged as a key metabolic regulator in many organs and tissues [3,4,5,6,7]. We and others have previously summarized ghrelin’s effects on glucose and energy homeostasis, adiposity, cardio-protection, muscle atrophy, bone metabolism, inflammation, and aging [8,9,10,11,12,13,14]. Here, we summarize ghrelin’s potential role as a modifier of cancer risk and progression.

The impact of ghrelin and its various isoforms is a growing field of research. Inconsistencies across studies have led to a body of literature that is somewhat difficult to interpret. Data from in vitro studies suggest that the effects of ghrelin may be both dose- and cell type-dependent, further complicating this field [15,16,17]. There are previous reviews that have summarized the role of ghrelin in specific cancers [18,19,20,21]; however, to our knowledge, no comprehensive review focusing on commonalities between the ghrelin axis across all cancers exists. Here, we discuss the roles of the ghrelin axis in tumor development and progression, highlighting the current state of the knowledge across the cancer spectrum, and identify areas important for future investigation.

## 2. Ghrelin Structure and Function

Ghrelin is encoded by the ghrelin prepropeptide gene (*GHRL*), which is translated to preproghrelin, a 117-amino-acid-long product [22]. Preproghrelin then undergoes alternative splicing, cleavage, and acylation as shown in Figure 1. Preproghrelin is cleaved to produce proghrelin, which yields the peptides ghrelin and obestatin, which have opposite functions in stimulating and suppressing appetite, respectively [12,22]. Proghrelin is acylated by the enzyme ghrelin O-acyltransferase (GOAT) and further cleaved to yield ghrelin [1]. If not acylated, proghrelin is cleaved to produce unacylated ghrelin (UAG). The *GHRL* gene contains six exons and four introns in total, and the introns enable alternative splicing [4] as shown in Figure 1. Two splice variants of particular interest are the In1-ghrelin variant and exon 3-deleted preproghrelin [23]. These splice variants share similar enough amino acid sequence homology with native ghrelin to permit recognition and acylation by GOAT [24].

It has been reported that the majority (>90%) of native ghrelin in circulation is found in the unacylated form, with only a small portion existing as acylated ghrelin (AG) [12,25]. However, there is some speculation that sample collection, processing and analytical technique may bias these results. For example, conversion of AG to UAG occurs via active proteases and/or acid hydrolysis, and a study by Blatnik et al. suggested that a significant portion of UAG measured in human plasma results from conversion after sample collection [26]. AG is the only form capable of binding to the ghrelin receptor growth hormone secretagogue receptor (GHSR), a G-protein coupled receptor, to augment the release of growth hormone (GH) and promote food intake [1]. Two variants of GHSR have been identified to date, known as type 1a and 1b (GHSR-1a and GHSR-1b). AG binds to GHSR-1a, while the function of GHSR-1b is still not completely understood [24].

## 3. Functional Role of Ghrelin Splice Variants across Tumor Types

Among the known ghrelin splice variants associated with cancer, the In1-ghrelin variant, which retains intron 1 but excludes exons 3 and 4 (Figure 1), is highly expressed in several cancers, including breast, prostate, pituitary, and neuroendocrine tumors [23,24,27,28,29]. This variant shares the same initial 13 amino acid sequence as native ghrelin, which includes the sequence required for acylation by GOAT and activation of GHSR-1a [24,30]. In high-grade breast tumors, an eightfold upregulation of In1-ghrelin has been observed compared to normal breast tissue [24]. In1-ghrelin is the only component of the ghrelin axis consistently elevated in pituitary adenomas [23]. Similarly, in high-risk prostate cancer patients, In1-ghrelin (but not native ghrelin) is significantly elevated in both plasma and tissue samples compared to that of normal controls [27].

In addition to being upregulated in several cancer types, In1-ghrelin is also associated with tumor aggressiveness [23,24,27,28,29]. Several studies have shown that In1-ghrelin levels positively correlate with expression of Ki67, a key cell proliferation marker [24,27,28]. Cancer cells transfected with In1-ghrelin also display higher rates of proliferation and increased ability to migrate and metastasize [24,27,28,29]. In addition, basal expression of In1-ghrelin is higher in the more invasive breast cancer cell line MDA-MB-231, compared to the less invasive cell line MCF-7 [29]. Finally, in a cohort of breast-cancer patients, high expression of In1-ghrelin was associated with lower-disease free survival compared to low or moderate expression [29]. Further, rates of metastasis were lower in patients with lower levels of In1-ghrelin, and follow up studies found that higher tumor expression of In1-ghrelin was associated with increased recurrence [29]. In1-ghrelin, therefore, may have potential to serve as a valuable biomarker, with the potential to provide information on the severity and aggressiveness of tumors.

Although the mechanisms through which In1-ghrelin acts are not fully understood, In1-ghrelin mRNA expression levels appear to strongly correlate with GOAT, as shown in Figure 2, which suggests that In1-ghrelin may be acylated similarly to native ghrelin [23,24,27,28]. Furthermore, there is evidence suggesting that In1-ghrelin can bind to and activate GHSR-1a, since addition of In1-ghrelin derived peptides to GHSR-1a transfected cells induced changes in calcium signaling, a relevant second messenger [23]. However, the absence of GHSR-1a receptor expression in prostate cancer cells suggests the potential for alternative unidentified receptors that mediate additional effects of In1-ghrelin [27]. Several studies have demonstrated that In1-ghrelin levels parallel GHSR-1b levels, but this association has not yet been explored in depth [24,28].

In addition to inducing an increase in intracellular calcium, In1-ghrelin treatment also increases phosphorylation of ERK, with no effect on Akt phosphorylation [27,29]. As seen in Figure 2, this suggests that In1-ghrelin has effects associated with the activation of MAPK/ERK signaling [23,27,29]. The MAPK/ERK pathway has been associated with cell proliferation, migration, and apoptosis; activation of this pathway could potentially explain many of the observed effects of In1-ghrelin [3,31]

Similar to the In1-ghrelin variant, the understudied exon 3-deleted ghrelin mRNA variant has been shown to be more highly expressed in malignant breast and prostate tissues compared to control tissues [32,33]. Exon 3 contains the gene responsible for production of obestatin [24]. There is limited data regarding the role of obestatin in these tumors. Obestatin does not appear to activate MAPK signaling, affect cell proliferation, nor influence apoptosis in prostate cancer cell lines [32,33]. However, its high expression in breast and prostate cancer cell lines suggests that it may have an important role that has yet to be discovered [32,33].

## 4. Acylated Ghrelin (AG) and Unacylated Ghrelin (UAG)

Previous research has implied that AG promotes tumor development, with low levels of AG leading to increased proliferation and inhibition of apoptosis in human ovarian cancer cells [15]. These proliferative effects of AG appear to be mediated through GHSR-1a, as treating cells with a GHSR-1a inhibitor blocks AG-induced proliferation [15] (Figure 2). Data from colon cancer cells have shown similar outcomes, where AG-induced proliferation has been shown to act through the GHSR-1a dependent Ras/PI3K/Akt/mTOR pathway [15]. In bladder cancer tissue, GHSR-1a is also highly expressed, and ghrelin induces immunosuppressive T regulatory cells, which would promote cancer cell proliferation [34]. Together, these studies suggest that circulating AG may initiate downstream effects in the body that ultimately lead to cancer proliferation.

When it comes to native ghrelin, in vitro studies are complicated by the range of doses (0.1 nM to 1000 nM) used, which may contribute to the diversity of responses reported both between and within different cell types. Across most cell types, low doses of ghrelin generally stimulate cancer cell proliferation; however, as summarized in Table 1, the results tend to be less consistent when higher doses of ghrelin are used. For example, in ovarian cancer cells, low concentrations of ghrelin prevent apoptosis; however, treatment with similar levels induces apoptosis in human colorectal adenocarcinoma cells [15,16]. Low to moderate doses of ghrelin have been shown to increase proliferation of endometrial cancer cells, but high levels show an inhibitory effect, possibly due to cytotoxicity [17]. High levels of acylated ghrelin have also been shown to induce apoptosis in ovarian cancer cells, but it is unclear if this is also due to cytotoxicity [15,35]. Higher concentrations of UAG and total ghrelin levels were associated with decreased risk of developing epithelial ovarian cancer in a cohort of post-menopausal women [36]. On the other hand, recently an in vitro study found that lower doses of ghrelin significantly increase cell proliferation of the A2870 ovarian cancer cells (Table 1). In clinical cohorts, decreased circulating ghrelin has been associated with increased risk of developing gastric cancer [37]. In vitro studies have similarly shown that overexpression of ghrelin in gastric cancer cell lines inhibited cell proliferation, migration, and invasion, and promoted apoptosis through the AMPK pathway, at least in one gastric cancer cell line (AGS cells) [38]. Conversely, ghrelin overexpression in the AGS and SGC7901 gastric cancer cells increased expression of the oncogene CDK6 and suppressed the expression of tumor suppressor gene p53 in the cell lines [39], suggesting a pro-tumorigenic effect.

The lack of standardization of experiments likely led to the conflicting results, which has made it difficult to assess the impact of ghrelin on tumor proliferation and growth, and difficult to identify a consistent effect across different tumor types. Instead, the effects of high doses of ghrelin most likely vary across tumor type, and the acylation state of ghrelin may also play an important role in the resulting effects. Examining how the ratio of UAG/AG affects cancer cell proliferation may yield interesting results in the future and allow better understanding of the underlying mechanisms involved. Overall, the interaction between AG, UAG, and various cancers is not well understood, and future studies need to be well controlled and use physiologically relevant ghrelin doses to shed light on this controversial aspect of ghrelin biology.

## 5. GHSR-1a/1b

Although GHSR-1a is well established as the primary receptor for ghrelin, the function of the truncated receptor GHSR-1b is not fully understood [24,28,42]. It is very intriguing that GHSR-1b, but not GHSR-1a, is overexpressed in many tumor types, including breast, neuroendocrine, adrenocortical tumors, prostate, and colorectal cell lines [24,28,32,42,47,48]. Studies have shown that GHSR-1b correlates with tumor grade/stage, as well as risk of metastasis [28,47] (Figure 2). Further, there also appears to be a positive correlation between tumor expression of the In1-ghrelin variant (discussed above) and GHSR-1b expression [24,28].

Despite a clear role in differentiating between a normal and cancerous state, the biological role and mechanism of the truncated GHSR-1b variant remains unknown [42]. Studies have shown that GHSR-1b is capable of heterodimerizing with GHSR-1a and promoting translocation, thereby blocking downstream signaling transduction; however, this functionality does not explain its role in tumor progression [28]. There are no known reports demonstrating that ghrelin is capable of binding GHSR-1b, yet the overexpression of GHSR-1b in tumors suggests that the proliferative effects of ghrelin might somehow be mediated through this receptor [32,47]. On the other hand, some studies report that the GHSR-1b receptor is inert in function, which could suggest the presence of an unknown GHSR subtype that shares enough sequence similarity to GHSR-1b to be misidentified by studies up to this point [47]. Together, these data suggest that GHSR-1a and -1b may have distinctive functions in cancer progression, and GHSR-1b may serve as potential clinical biomarker in cancers.

## 6. GOAT

GOAT, the enzyme involved in the acylation of ghrelin, was identified in 2008 [49]. This enzyme may also be involved in cancer development. While not overexpressed as frequently as GHSR-1b, GOAT overexpression has been identified in several cancers, including breast, prostate, and neuroendocrine tumors [24,27,28,50,51,52]. Though GOAT is responsible for acylating ghrelin to produce its active signaling form, GOAT expression does not appear to correlate with expression of native ghrelin in any tumor types studied [24,27,28,50]. On the other hand, a positive correlation between GOAT and the In1-ghrelin variant has been observed, suggesting that In1-ghrelin (rather than native ghrelin) may actually be the predominant substrate for GOAT in cancer cell pathology [24,27,28,30,50,52]. This relationship only further reinforces the idea that In1-ghrelin could serve as a signaling peptide, making this an exciting area for future investigation.

Several studies have explored the potential role for GOAT as a biomarker in patients with prostate cancer [51,52,53]. GOAT was found to outperform prostate-specific antigen (PSA) in the diagnosis of intermediate to high-grade prostate cancer [51]. PSA is a highly valuable biomarker of prostate cancer risk; however, there is a “grey zone” for PSA, where a slight elevation in PSA is not a reliable marker of prostate cancer; thus, the potential application of GOAT as a biomarker for prostate cancer diagnosis is very exciting [30,51,52,53]. Plasma and urine GOAT levels were found to correlate with presence of metastases and molecular markers of tumor aggressiveness, suggesting that this may be a non-invasive means of predicting prostate cancer aggressiveness [51,52,53]. GOAT also has potential as a therapeutic target, because data have shown that prostate cancer cells secrete GOAT, GOAT overexpression increased prostate cancer cell proliferation, and silencing GOAT had the opposite effect [52,53].

## 7. Ghrelin-Related Signaling Pathways

Ghrelin and unacylated ghrelin (UAG) have both demonstrated the ability to stimulate cancer cell proliferation [54], and this is thought to occur primarily through activation of the PI3K/Akt pathway [15,45,46,54,55,56,57,58,59]. Like the MAPK/ERK pathway, the PI3K/Akt pathway plays a significant role in regulating cell proliferation, apoptosis, metabolism, and angiogenesis [60]. Acylated ghrelin has been shown to induce phosphorylation of PI3K and Akt in a GHSR-1a dependent manner in ovarian cancer cells, breast cancer cells, non-small cell lung cancer cells, and pancreatic adenocarcinoma cells [15,55,56,57]. Treatment with Akt and PI3K inhibitors appears to inhibit ghrelin-induced proliferation in cancer cell lines in several studies [45,54,56,57]. In studies where ghrelin-induced proliferation was only partially blocked by inhibitors of the Akt/PI3K signaling pathway, near-complete inhibition was accomplished through simultaneous addition of an ERK1/2 inhibitor [54,56]. GHSR inhibitors also block ghrelin-related activation of the PI3K/Akt pathway and downstream proliferative effects in vitro, which suggests a reliance on GHSR activation for this pathway [15,55,56]. Like acylated ghrelin, UAG is also able to stimulate Akt phosphorylation and cell proliferation in gastric adenocarcinoma and pancreatic beta cells [54,61]. Since UAG is unable to bind to GHSR-1a, this also supports the notion that an alternative unknown receptor may mediate some of ghrelin’s effects [54,61] (Figure 2).

Emerging in vitro data has also shown that ghrelin may modify response to cancer therapeutics, specifically the response to the chemotherapy agent cisplatin. Cisplatin decreases phosphorylation/activation of PI3K, Akt, and mTOR, leading to decreased cell proliferation [55]. However, treatment with acylated ghrelin rendered both breast and ovarian cancer cell lines more resistant to cisplatin’s apoptotic effects [15,55]. Use of the PI3K inhibitor LY294002 and mTOR inhibitor rapamycin reduced the anti-apoptotic effects of ghrelin, as did the addition of GHSR siRNA, suggesting that ghrelin’s pro-survival effects are mediated through GHSR-1a and PI3K/Akt activation [55]. It should be noted that these effects were observed only under low doses of ghrelin [8,25,26,27,28,29]. In human non-small cell lung cancer, inhibition of the PI3K/Akt pathway by the mTOR inhibitor, rapamycin, proved to cease cell proliferation and ghrelin-induced phosphorylation [56]. Additionally, pretreatment with rapamycin led to a significant reduction of ghrelin-induced proliferation in colon and breast cancer cells [46,55], and LY294002 and PD98059 (an inhibitor of MAPK), have independently demonstrated a role in partially blocking ghrelin-induced proliferation [56]. Interestingly, LY294002 combined with PD98059 completely attenuated cell proliferation induced by ghrelin in A549 cells, thus suggesting a possible synergistic role in inhibiting ghrelin’s proliferative effects [56].

The role of ghrelin on cyclooxygenase-2 (COX-2) shows conflicting results, but is an area of interest in the cancer and ghrelin relationship [41,44,62]. COX-2 has been linked to cancer initiation, progression, and metastasis [63,64,65,66,67,68]. Many of the effects of COX-2 are attributed to the downstream metabolite prostaglandin E2 (PGE2), originating from arachidonic acid [63,64]. Li et al. [44] demonstrated that gastric cancer cells treated with ghrelin had increased COX-2 expression, and this associated with decreased apoptosis and increased invasive potential (Table 1). These effects were blunted and COX-2 protein expression decreased when cells were treated with an Akt inhibitor, suggesting that ghrelin uses the PI3K/Akt pathway to upregulate COX-2 [44]. Konturek et al., however, found that ghrelin decreased TNF alpha-induced COX 2 expression in an esophageal adenocarcinoma cell line [41]. Ghrelin has also demonstrated the ability to upregulate arachidonic acid release through enhancing phospholipase A2 activation [69,70,71]. Therefore, ghrelin can increase both intracellular arachidonic acid concentration and COX-2 expression via PI3k/Akt, leading to increased pro-tumoral PGE2 synthesis, which may enhance tumor proliferation, survival, and progression. Given the conflicting findings in this field, the role of ghrelin in the modulation of COX-2 and downstream effects on tumors is an area ripe for further investigation.

## 8. Ghrelin in the Prevention of Cancer Cachexia

In addition to evaluating ghrelin’s role in cancer cell proliferation, an increasing number of studies have attempted to determine ghrelin’s role in cancer cachexia. Cachexia, a complex muscle wasting syndrome, remains a major issue in patients with advanced cancers [34,58,59]. A recent study investigating the therapeutic effects of AG and UAG in a murine model found that administration of AG and UAG improved nutritional status and partially reversed the effects of cachectic muscle wasting [72]. Interestingly, when AG and UAG were administered to the mice, the activation of myofilament degradation processes were inhibited [72], suggesting that AG and UAG have the potential to ameliorate cancer cachexia. In individuals with cancer cachexia undergoing chemoradiotherapy, ghrelin levels and overall body weights appeared to significantly decline [73]. This suggests that during such treatments, therapies targeting ghrelin levels may be beneficial in decreasing the negative effects of cachexia.

In order to find additional approaches to improve symptoms of cancer cachexia, growth hormone secretagogues, also known as ghrelin receptor agonists, have been investigated. These ghrelin receptor agonists increase body weight, lean body mass, and food intake [74,75,76], which would be beneficial for patients with cachexia. Over the past decade, one specific ghrelin receptor agonist, anamorelin, has emerged as an agent of interest. Anamorelin is an orally active and selective agonist of the ghrelin receptor that has recently been approved for use [76,77]. Consistent with previous studies, this drug was shown to improve body weight, lean body mass, and appetite in a randomized, double-blind study in patients with cancer cachexia [78]. However, anamorelin did not appear capable of improving motor function or overall survival of these patients [78]. Additional studies have similarly reported significant increases in lean body mass with administration of anamorelin [74,76,77]. However, when measuring handgrip strength following anamorelin administration, no significant improvements could be noted [74,76,77,78]. Despite the somewhat disappointing lack of effect on muscle strength and motor function, anamorelin remains a potential treatment option to ameliorate the other aspects of cancer cachexia.

When considering the use of anamorelin in attenuating cachexia, potential increases in circulating GH and insulin-like growth factor (IGF-1) are of concern. Whether elevated levels of these growth factors would be sufficient to drive tumor development and growth is still a matter of debate [79,80]. Few studies have looked closely at the relationship between the ghrelin axis, GH, and IGF-1 in relation to tumor progression, but one such study involving a mouse model of non-small cell lung cancer found no difference in tumor growth between mice treated with ghrelin or anamorelin and untreated controls, despite slightly higher plasma IGF-1 and peak GH levels in the treated animals [81]. Further studies examining the relationship between ghrelin, the GH/IGF-1 axis, and tumor growth are needed to clarify these interactions.

## 9. Ghrelin, Obesity, and Cancer Feedback Loop

Recent research has also identified the ghrelin axis as a potential link between obesity and risk of hormone-related cancers, including both prostate and post-menopausal breast cancer.

Specifically, for prostate cancer, increased expression of the GOAT enzyme was found in prostate cancer patients with overweight/obesity status, dyslipidemia, and/or diabetes [52]. Urinary levels of In1-ghrelin were also elevated in obese and diabetic prostate cancer patients compared to non-diabetic patients of normal weight. Further, urinary GOAT levels in these patients also positively correlated with circulating levels of glucose, insulin, triglycerides, and glycated hemoglobin [30]. These factors are common markers of obesity and insulin resistance, and are associated with increased risk of prostate cancer and prostate cancer aggressiveness [30]. High urinary levels of In-1ghrelin are also independently associated with increased risk of prostate cancer. Together these findings suggest that dysregulation of In1-ghrelin signaling could link an impaired metabolic state to prostate cancer [30].

Ghrelin may also serve as a critical link between obesity and post-menopausal breast cancer, affecting not only systemic metabolism, but also production of the tumor-promotional estrogens [19]. Ghrelin levels are known to inversely correlate with measures of obesity, including BMI, percent body fat, and circulating levels of both insulin and leptin [82,83]. It has been shown that ghrelin’s effect on obesity in mouse models is dependent on diet and genetic background. Suppressing ghrelin exacerbates fructose diet-induced adiposity and insulin resistance [84], but has no effect on leptin-deficiency-associated obesity while improving hyperglycemia [85]. In addition, both AG and UAG have been found to be potent inhibitors of aromatase expression and activity in adipose stromal cells (ASCs) [83]. Aromatase is the enzyme responsible for the conversion of androgens to estrogens, and its expression by ASCs is considered an important factor in the development and growth of estrogen-dependent cancers after menopause [83,86]. The mechanism appears to be GHSR-1a independent, and possibly caused by a decrease in intracellular cAMP levels [83]. UAG was also found to inhibit the ability of macrophages to activate aromatase expression in ASCs [86]. This discovery explores a possible explanation of the link between obesity and post-menopausal breast cancer, as lean women would likely have higher levels of ghrelin that would keep adipose-produced aromatase levels much lower, reducing the severity of estrogen-dependent cancers in these patients [83,86]. Since AG stimulates proliferation of MDA-MB-231 cells, a triple-negative cell line, estrogen receptor status could be a factor responsible for the mixed reactions of breast cancer cell lines to ghrelin [32].

## 10. Interactions between Ghrelin and Other Hormones: Implications for Cancer

Ghrelin is known to interact with many other hormones; however, the impact of these interactions with respect to cancer have not been studied in great detail. The adipokine leptin is of particular interest; it is secreted from adipose tissue and plays key roles in regulation of energy balance [87]. Leptin and ghrelin target similar regions within the hypothalamus and brain stem in order to exert opposing effects—anorexigenic and orexigenic effects, respectively [88]. Individuals with obesity tend to have lower levels of circulating ghrelin and higher levels of circulating leptin [89]. Leptin exerts its effects through activation of the leptin receptor ObR, of which the most predominant isoform is ObRb [88]. Upon binding, leptin recruits Janus Kinase (JAK) and typically initiates cascades involving STAT3 and PI3K signaling [88]. Both direct and indirect effects of leptin in regulating cancer proliferation, metastasis, angiogenesis, and chemoresistance have been identified [90], and leptin—either alone or in combination with other adipokines and cytokines—has been identified as a biomarker of cancer risk [91,92]. These pro-tumorigenic effects occur across several cancer types, through JAK-STAT, PI3K/AKT, MAPK/ERK, and COX-2 dependent pathways [93,94,95,96,97]. Considering the similarities in the mechanisms between which leptin and ghrelin exert their effects on cancer cells as well as their observed inverse relationship, the interactions between ghrelin and leptin could prove an interesting point for future study in relation to cancer.

Given the significant role of ghrelin in maintaining metabolic homeostasis, it comes as no surprise that insulin has known interactions with ghrelin [8]. Ghrelin is now recognized as a major regulator of glucose homeostasis [10] and plays an important role in the pathogenesis of diabetes [85]. Ghrelin suppresses insulin secretion by regulating uncoupling protein 2 (UCP2) in panarctic β cells; ghrelin ablation attenuates hyperglycemia of leptin-deficient *ob/ob* mice without affecting obesity [85]. While there is some disagreement in the exact interactions between these two hormones, there are patterns that have been identified across studies. Infusions of AG have been shown to suppress early insulin and glucose responses, while UAG infusion was found to enhance early insulin response and reduce circulating free fatty acid (FFA) levels, countering the effect of AG [98,99,100]. On the other hand, combined administration of AG and UAG significantly improved insulin sensitivity compared to placebo [99]. UAG was further observed to reduce infiltration of macrophages into adipose tissue, and also promoted macrophage polarization from the proinflammatory M1 subtype to the anti-inflammatory M2 subtype, a shift that has been correlated with improvements in insulin sensitivity [101]. These results demonstrate that the acylation state of ghrelin plays a significant role in its interaction with insulin. Given the known role that insulin resistance and adipose inflammation play in the development and progression of cancers, particularly obesity-associated cancers [102,103,104,105,106], this provides another possible link between ghrelin, insulin signaling, inflammation, and cancer.

Liver-expressed antimicrobial peptide 2 (LEAP-2) is a key endocrine factor in systematic energy metabolism [107], which also interacts with the ghrelin system. LEAP-2 acts as an endogenous noncompetitive allosteric antagonist for GHSR-1a, impairing ghrelin’s ability to activate its receptor [107]. LEAP-2 can help prevent ghrelin-induced adiposity making it a potential therapeutic for obesity [107,108]. To our knowledge, there are no studies that focus on the relationship between LEAP-2, GHSR-1a, and cancer, making it a potentially interesting area of research, especially concerning the ghrelin, obesity, cancer feedback loop.

The role of ghrelin in the hypothalamic–pituitary–gonadal (HPG) axis is another recent topic of interest. Ghrelin has been observed to inhibit the release of gonadotropin-releasing hormone (GnRH) in ovariectomized rats in addition to also reducing the responsiveness of luteinizing hormone (LH) to GnRH in prepubertal male rats and gonadectomized rats [109,110]. Little to no effect was seen on the release of follicle-stimulating hormone (FSH), and the mechanism by which ghrelin selectively modulates LH release is still unknown [109]. Considering the wealth of existing information regarding GnRH agonists and their potential for reducing tumor cell proliferation, ghrelin’s unique relationship with GnRH could yield further insight into the relationship between ghrelin and reproductive cancers [111]. Further confirmation of this relationship in a tumorigenic setting would be necessary before any connections are hypothesized.

## 11. Strengths and Limitations

While our knowledge of the link between the ghrelin axis and cancer continues to expand, much of the concrete data presented in this review come from in vitro studies in established cell lines. As such, extrapolating these findings to a clinical setting must be conducted with caution. Additional studies using in vivo models and translation to clinical populations will be necessary moving forward. This field has many areas that warrant further investigation, including a need to move beyond association studies to well-controlled evidence-based interventional studies to identify cause–effect relationships and to further understand the underpinning mechanisms.

Despite these limitations, this review provides a narrative review based on an extensive review of the literature, and has identified many areas for future research. We have highlighted several areas of uncertainty and conflict in the field, and hope this provides a platform for the design of future studies to address these gaps in our current knowledge.

## 12. Conclusions

Although there are many unanswered questions, including more than could be fully described in this review, strong associations have clearly been established between ghrelin and tumor development/progression across several tumor types. Specifically, the In1-ghrelin variant, the enzyme GOAT, and the GHSR-1b receptor appear to play key roles in tumor proliferation and cancer aggressiveness, as summarized in Figure 3. The role of native ghrelin in cancer is not fully understood, but studies suggest that effects are dependent on cell type and levels of this hormone; native ghrelin likely has proliferative effects at low doses through the activation of the PI3K/Akt pathway (Figure 2 and Figure 3). While the mechanisms of action require further investigation, current data supports the notion that many forms of ghrelin have the potential to serve as biomarkers of cancer risk and progression and suggest that the ghrelin family could also be explored as a target for cancer treatment.

In1-ghrelin holds strong potential as both a biomarker and a molecule to be studied in greater depth to better understand its role in cancer cell proliferation, especially considering its possible roles in signal transduction pathways (Figure 3). The potential role of UAG also holds great interest, especially considering that UAG was thought to be an inactive form of ghrelin, yet it is capable of activating proliferation-related signaling pathways. This suggests the presence of an alternate unidentified receptor that may mediate these effects (Figure 2), a hypothesis aided by the enigma of the GHSR-1b receptor. Despite GHSR-1b being considered nonfunctional, its frequent overexpression in cancer tissues suggests that there might be an alternate functional receptor, with similar sequences. Further, the exon-3 ghrelin variant is overexpressed in some hormonally related cancers, but little is known about this variant; thus, this may merit further study. Together, literature suggests that ghrelin axis has important roles in tumor development and progression. Further investigation into how the ghrelin axis drives tumor development and growth may lead to novel approaches to identifying/treating different forms of cancers.

## Figures and Tables

**Figure 1 biomolecules-12-00483-f001:**
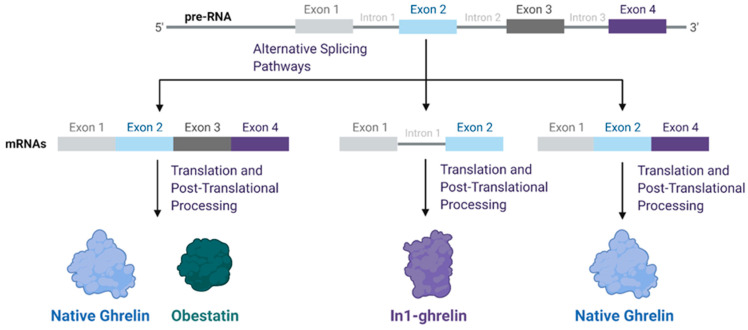
Gene splicing of ghrelin and splice variants. Diagram summarizing the different splicing pathways leading to the generation of the various mRNA transcripts formed from *GHRL* transcription. Splicing of exons 1–4 yields products of native ghrelin (acylated or unacylated) and obestatin. The inclusion of intron 1 coupled with the exclusion of exons 3–4 leads to the production of the In1-ghrelin variant. Exclusion of exon 3 alone yields the exon-3 deleted preproghrelin transcript, which is further processed to produce native ghrelin.

**Figure 2 biomolecules-12-00483-f002:**
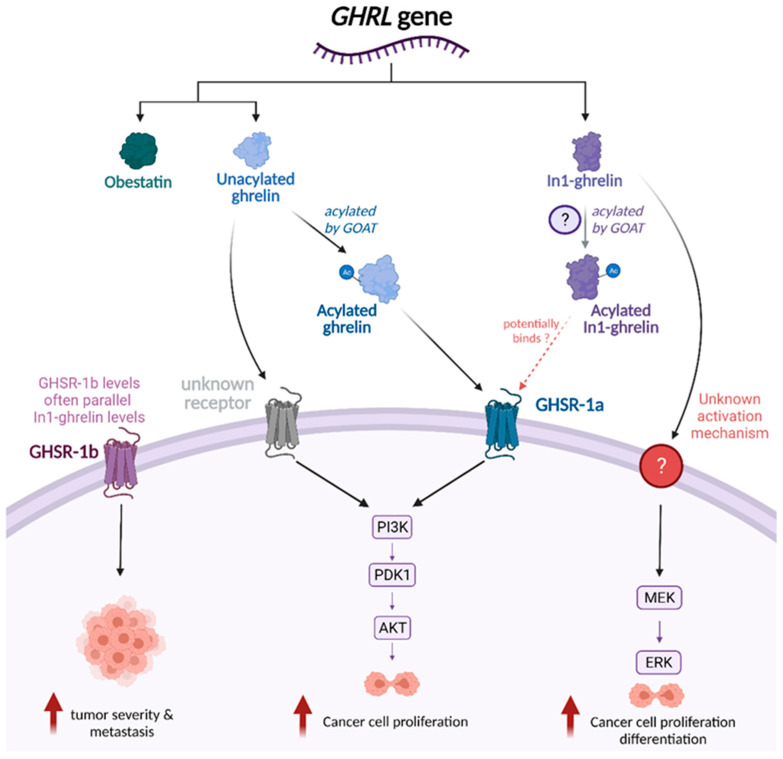
Proposed mechanistic links between ghrelin axis and tumor growth and metastasis. Proposed model illustrating pathways through which the ghrelin axis exerts proliferative effects on cancer cells. UAG and AG have both been found to activate the PI3K/Akt pathway and stimulate cancer cell proliferation in low doses, though AG acts in a GHSR-1a dependent manner while UAG’s mechanism of activation is unknown. In1-ghrelin has been found to activate the MAPK/ERK pathway through an undiscovered mechanism. In1-ghrelin levels have also been found to correlate with GOAT levels, suggesting that acylation and corresponding activation of GHSR-1a may be possible, though In1-ghrelin has not been reported to alter Akt phosphorylation. GHSR-1b levels often parallel In1-ghrelin levels as well, and have been linked to increased tumor severity and metastasis.

**Figure 3 biomolecules-12-00483-f003:**
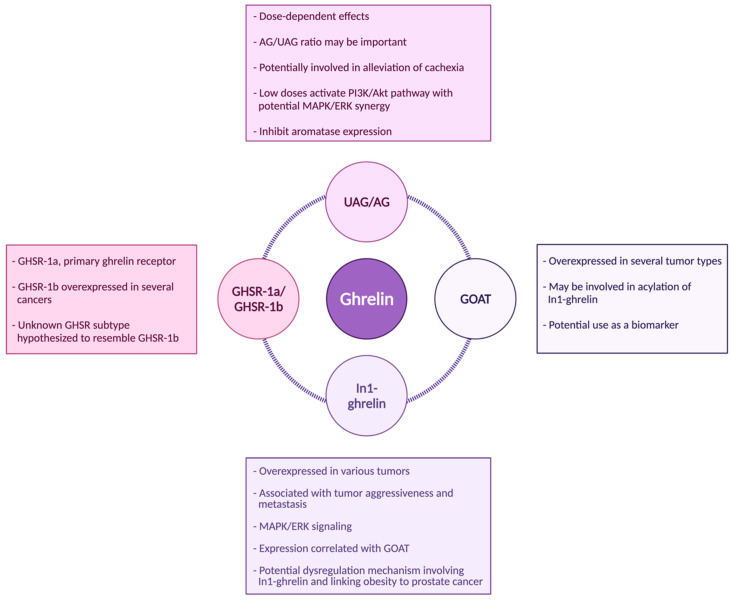
Summary of the proposed relationship between the ghrelin axis and cancer, highlighting the known and proposed roles of UAG, AG, In1-ghrelin, GHSR-1a, GHSR-1b, and GOAT.

**Table 1 biomolecules-12-00483-t001:** Summary of in vitro studies of ghrelin across cancer types, including dose, acylation status of ghrelin, and effects on cell proliferation and survival. Arrows represent directionality: increased/enhanced (↑) or decreased/suppressed (↓).

Cancer Type	Cell Line	Ghrelin Acylation Status	Concentration	Effect	Reference
Breast	MCF7 (ER+/PR+/HER2-)	Acylated	0–1000 nM	No effect	[32]
	SKBR3 (HER2+)	Unacylated	0.1 nM	Inhibited growth	[40]
	MDA-MB-435 (TNBC)	Acylated	0.1, 1, 10, 100 nM	↑ proliferation	[32]
	MDA-MB-231 (TNBC)	Acylated	10 and 100 nM	↑ proliferation	[32]
Ovarian	A2780	Acylated	1 nM	↑ proliferation	[15]
Endometrial	HEC1B	Acylated	10 and 100 nM	↑ proliferation	[17]
	KLE	Acylated	1, 10, 100 nM	↑ proliferation	[17]
Esophageal	OE-19	Not Specified	20–450 nM	No effect on apoptosis	[41]
Prostate	PC3	Acylated	5 and 10 nM	↑ proliferation	[42]
	DU145	Unacylated	100 and 10,000 nM	Inhibited growth	[43]
		Acylated	10–1000 nM	Inhibited growth	[43]
Gastric	AGS	Not Specified	Overexpression	↑ apoptosis	[38]
		Not Specified	10 and 100 nM	↑ proliferation	[39]
		Not Specified	10 nM	↑ cell migration and invasion ↓ apoptosis	[44]
	SGC7901	Not Specified	1, 10, 100 nM	↑ proliferation	[39]
Colon	HT-29	Acylated	0.1 and 1 nM	↑ proliferation	[45]
	Caco-2	Acylated	1 and 10 nM	↑ proliferation	[46]
		Unacylated	1 and 10 nM	↑ proliferation	[46]

## Data Availability

No new data were created or analyzed in this study. Data sharing is not applicable to this article.

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
