# Peer review of "Ghrelin and Cancer: Examining the Roles of the Ghrelin Axis in Tumor Growth and Progression"

_biomolecules, 2022, doi:10.3390/biom12040483_

Round 1

Reviewer 1 Report

This review is very interesting. The subject of this review is original, well written and easy to read. In terms of the organisation of this review, the authors provide concluding remarks that answer the question "does ghrelin have a key role in the progression of certain types of cancer" and "what are the main cell signalling pathways involved ?

On the other hand, this review needs to be completed on the part "7. Ghrelin-related Signaling Pathways". Indeed, a paragraph on ghrelin/arachidonic acid metabolism relationships should appear in this section. Several studies have shown relationships with phospholipase A2 activity followed by modulation of cyclooxygenase-2 expression representing a cell survival pathway in many types of cancer (i.e. "Ghrelin Regulates Cyclooxygenase-2 Expression and Promotes Gastric Cancer Cell Progression", DOI: 10.1155/2021/5576808).

Author Response

We thank the reviewer for their positive feedback and constructive comments. As suggested a paragraph on ghrelin/arachidonic acid/COX-2 has been added to the revised manuscript, and we think this has made for a stronger paper.

Reviewer 2 Report

The authors reported a review summarizing ghrelin's potential role as a modifier of cancer risk and progression, mainly based on basic research articles. The topic seems to be a unique area of great interest to readers. The review process was not systematic but narrative. However, their choices and classifications of references on this topic were reasonable. Knowns and unknowns regarding this specific topic were discussed appropriately and would support future studies. However, the indirect effect of ghrelin on tumorigenesis and progression and feedback loop was not fully described. Accordingly, I think this excellent manuscript needs the following revisions before acceptance. Thanks.

Comments

  1. The role of GH and IGF-1 induced by ghrelin or GHSR-1a agonist may be important to enhance tumorigenesis, progression, and aggressiveness. I know this is an unresolved issue, but I think it is an important part of the ghrelin axis when discussing its potential role in tumor growth. Please summarize it.
  2. The interactions with other peptide hormones, especially the endogenous agonists, need to be discussed more in the feedback loop section. Liver-expressed antimicrobial peptide 2, leptin, insulin, somatostatin, catecholamines, cholecystokinin, and others may be included.
  3. Please describe the strength and limitations of this review before the conclusion.

Round 2

Reviewer 1 Report

On pages 7-8, the sentence corresponding to lines 278-282 should be replaced by the following: « Ghrelin can increase both intracellular arachidonic acid concentration and COX-2 expression via PI3k/Akt, leading to increased pro-tumour PGE2 synthesis, which may enhance tumour proliferation, survival and progression. »

Author Response

The sentence has been modified as suggested. Thank-you for recommending this change.

Reviewer 2 Report

Dear Authors,
Thank you for your revised manuscript. I think that all the comments are responded to and adequately reflected in the revised manuscript. The authors discussed all biases and weaknesses. I think this manuscript is now acceptable. Thank you for all your efforts.

Author Response

No changes needed. Thanks!